# Mental and Physical Wellbeing of Carer–Employees in Canada

**DOI:** 10.3390/ijerph21121611

**Published:** 2024-11-30

**Authors:** Ito Peng

**Affiliations:** Department of Sociology, Munk School of Global Affairs and Public Policy, University of Toronto, Toronto, ON M5S 1X6, Canada; ito.peng@utoronto.ca

**Keywords:** carer-employees, caregiving, unpaid care, work-care tension, senior care, long-term care, aging population, family

## Abstract

Today, an increasing number of Canadian adults are providing unpaid care to their family members and friends while working full or part-time. We conducted a national survey of unpaid caregiving for older people in 2022 to learn who these people are, why they care, and to identify the social, economic, and health impacts of unpaid caregiving. Our findings show that many of these caregivers are also employees. While most research shows that women caregivers and carer-employees experience work-care tension that results in negative mental and physical health outcomes, our results are more mixed. This paper provides descriptive findings of carer-employee’s health and well-being, and compares them to previous research in Canada and abroad. I conclude with policy options for governments and employers to support the health and well-being of carer-employees.

## 1. Introduction

Shifting demographics are creating a growth in carer–employees—the population of workers who simultaneously engage in family caregiving. Key trends affecting this growth include: population aging, fertility decline, family distanciation, and the normalization of social and cultural expectations regarding adult women and men’s employment. These changes are having, and will continue to have, huge impacts on society, the economy, politics, and values. As the share of the global population over the age of 65 continues to increase, we can expect a heightened demand for senior and long-term care (The proportion of the global population aged 65+ is projected to increase from 10% in 2021 to 17% in 2050, with its number more than doubling from 761 million to 1.6 billion during the same time period) [1]. Indeed, already amongst the estimated 349 million people worldwide requiring care, 101 million are aged 60 years and older [2]. In Canada, people over the age of 65 made up 19.3% (7.6 million) of the total population in 2023; this figure is expected to increase to 25% by 2043 [3]. One consequence of these shifting demographics is that more workers will engage in family caregiving. In 2012, carer–employees (CEs) accounted for approximately 35% of Canada’s workforce (roughly 5.6 million), making up the majority of unpaid caregivers [4] (In a separate calculation based on the 2018 General Social Survey on Caregiving and Care Receiving, Magnaye et al. [5] claimed that 26.5% of Canadians (5.2 million) of employment age (19 to 70) were CEs. The differences in the proportions and numbers of CEs in 2012 and 2018 were due to the different definitions of employment used in the two surveys). Not only has there been a rise in the absolute number of CEs over the last decade, but the intensity of care has also risen. According to Wu et al., CEs spent more hours in caring per week on average in 2018 (10.3 h) than in 2012 (7.6); at the same time, their life satisfaction declined [6].

Given the trend toward more CEs in the future, and CEs’ unique role at the crossroads of work obligations and caregiving duties, families, employers, and policymakers need to better understand their distinctive socio-economic positions and the risks to their physical and mental health well-being. We conducted a quantitative national survey of unpaid caregiving for older people in Canada in 2022 to learn who these people are, why they care, and to identify the social, economic, and health impacts that they experience from unpaid caregiving. This paper discusses the findings from the survey, focusing on the health and well-being outcomes, comparing them with previous research in Canada and abroad. I conclude with a discussion of the policy implications of our findings.

## 2. Theoretical/Conceptual Explanations of the Impacts of Unpaid Caregiving on Carer-Employee Health and Well-Being

Research shows that juggling work and caregiving has direct and indirect impacts on employees’ work, health, and well-being. This is particularly relevant for women, not only because women are more likely than men to provide care, but also because their caregiving tends to be more direct (e.g., feeding, bathing, giving medications as opposed to providing transportation, managing care services, etc.) and more intense (more hands-on and involving direct physical and emotional care). Caregiving can affect worker performance and productivity by causing work interruptions, decreased performance, and increased absenteeism [7,8,9,10]. Additionally, research has found that combining employment with caregiving can result in negative physical and mental health outcomes, including increased stress, higher depression, and anxiety [6,11,12,13,14,15,16].

Researchers have applied several theoretical and conceptual frameworks to explain carer–employee (CE) health and well-being. The Conservation of Resource theory argues that caregiving poses a threat to an individual’s resources by depleting their time, energy, and other resources, which in turn creates or magnifies inter-role conflicts. For example, work–family conflict (WFC) and family–work conflict (FWC) can result in increased stress and other negative physical and mental health outcomes [17,18]. The combination of work and caregiving can create significant role conflicts as employees struggle to meet the demands of both roles within limited time and energy capacities. Numerous studies show that inter-role conflicts resulting from care burden have direct and negative impacts on CEs’ mental health [19,20,21]. Li et al., for example, found that higher levels of caregiving demands are strongly associated with FWC, whereby employees experience significant strain from family care responsibilities, resulting in decreased life satisfaction and increased depression [10]. Applying the Conservation of Resource theory and inter-role strains, Kayaalp et al.’s analysis also shows that both role-strain-based WFC and FWC are key contributors to the deterioration of mental health of CEs over time. They argue that “the emotional or psychological aspects of balancing work and family… is most important for mental health, particularly depressive symptoms, rather than the degree of time involved in balancing work and family responsibilities.” (2021:232) [22].

Dugan et al., however, counter Kayaalp et al.’s analysis of the non-temporal impacts of WFC and FWC (Kayaalp et al.’s analysis may be partially explained by the fact that their sample consisted mainly of CEs providing childcare rather than eldercare) [9]. They argue that, in the case of eldercare, CEs who are providing five or more hours per week of care experience higher FWC and increased depressive symptoms compared with CEs who are not providing eldercare or are providing less than 5 h of care per week. A recent study by Zhang and Bennett concurs with Dugan et al.’s analysis. Using longitudinal data from 1991 to 2017, their analysis found that the threshold or the “tipping point” at which care intensity begins to negatively impact caregivers’ psychological well-being is 5 h a week for those providing care to their parents [23]. They also found that the tipping point differs depending on the caregiver–care recipient relationship. For example, for people caring for their spouse, any level of care intensity as expressed in terms of numbers of hours per week is associated with negative psychological well-being, whereas for those caring for a child, the tipping point is 50 h. Contrary to some studies that show lower psychological well-being among parents caring for a child, Zhang and Bennett’s study thus argues that, when care hours are controlled, caregivers providing care to a child have higher well-being than those caring for their spouse or parent [23].

Building on the Role–Stress theory, Boumans and Dorant’s cross-sectional comparative study of caregivers’ psychological strain shows that work–home interference (WHI) and home–work interference (HWI) are significant determinants of CEs’ emotional and mental health [24,25]. Here, WHI refers to when work obligations negatively affect ECs personal lives, such as hindering participation in family activities due to work schedules, while HWI refers to when issues at home interfere with work performance, like difficulty concentrating at work due to domestic problems. Although both WHI and HWI highlight the conflict caused by incompatible demands from work and family domains, and they often result in emotional exhaustion and poorer mental health, this does not always have to be so; rather, depending on the situation, they can be beneficial. For example, responsibilities at home that align with work demands can create positive WHI, and efficiently organizing work time can lead to positive HWI.

The effort–recovery (E-R) model explains that, whereas excessive job demands can inhibit sufficient recovery at home, thus potentially harming health and well-being, when effort expenditure is manageable and aligns with recovery needs, positive work reactions, such as job satisfaction and motivation, can occur. Boumans and Dorant show that, while providing more hours of informal care is associated with lower mental and physical health, a more negative WHI and HWI, and a higher need for recovery at home, emotional exhaustion, and presenteeism, CEs rate their motivation and satisfaction as equally high as those of their non-CE colleagues and show no differences in terms of absenteeism [24]. Their subsequent analysis shows more specifically that caregiver strain is more strongly associated with job demands (workload, WFC) and family demands (care hours and FWC) than it is with job and family resources. Their study thus suggests that the effectiveness of aligning demands with recovery needs can make a difference in positive or negative WHI/HWI, and in influencing CEs’ experience of caregiving strain and mental health [25].

Using Life Course Theory, Chesley and Moen argue that, as one role (employment) shapes the context for another (caregiving), workplace context serves as an important basis for couples to make decisions about the division of their caregiving roles, and this can help to determine their health and well-being outcomes [11]. Using longitudinal data from The Ecology of Carer Study, their research shows that strategic decisions made by dual-earner CE couples are more likely to enhance the psychological well-being of the husband in the long run while detracting from the wife’s.

Wang et al. employed Pearlin et al.’s Stress Process model to understand caregivers’ stress and its impact on their physical and mental well-being [4,26]. The Stress Process model identifies four key areas: background and context factors (gender, age, race and other SES characteristics, family network, etc.); primary stressors (both objective and subjective indicators, such as the care recipient’s level of care needs, caregiver’s overall caregiving responsibilities, etc.); secondary stressors (WFC/FWC, economic problems, caregiver’s sense of self-esteem, sense of mastery, and other situational contexts); and mediating factors (social support and coping strategies). The outcomes considered cover both physical health concerns, like illness or injury, and mental health issues, such as depression, anxiety, and irritability. Their analysis, based on Statistics Canada’s General Social Survey (Cycle 26-2012) data, suggests that flexible work schedules are not necessarily beneficial. Indeed, CEs who have access to flexible work arrangements and experience specific work interruptions, such as reduced hours or taking days off, are less likely than CEs on regular work schedules with no interruptions to report unfavorable physical and mental health. This remains true in the case of mental health even for CEs providing over 10 h of care weekly. In a follow-up analysis, Wu et al. compared data from the Canadian General Social Survey in 2012 and 2018 [6]. By 2018, the proportion of male CEs had increased such that women were no longer more likely to be a CE than men; however, the intensity of caregiving had increased significantly more for female CEs than for their male counterparts, and general health and life satisfaction had significantly worsened for female CEs, but not for male CEs.

## 3. Materials and Methods

We undertook a national survey of unpaid caregiving for older adults (aged 65 and over) in July and August 2022. The survey was approved by the University of Toronto’s ethics board (Protocol #42856) during the spring of 2022. We conducted an online survey of adults over the age of 18 who identified themselves as the main caregiver of an adult aged 65 and over. The survey was conducted online because strict COVID restrictions during the spring of 2022 made it impossible to undertake in-person interviews. The survey questionnaire focused on 5 thematic areas: (1) household/family configurations: we asked questions about household size, gender and ages of people who lived in the household, their relationships with each other, whether the caregiver was living with the care receiver, etc.; (2) care arrangements and conditions: we asked questions about why and how the caregiver became a caregiver, their caregiving conditions, the social, economic, and health impacts (emotional, psychological and physical) of their unpaid caregiving, and what kind of help or support they received from their families/households and from whom; (3) decision-making: we were interested in learning how and when decisions were made about unpaid caregiving and its arrangements, who was involved in the decision-making and how (including whether the care recipients were involved), and what factors were involved in their decision-making; (4) time and form: we asked about how much unpaid care was given to an older person both in terms of time and types/forms of care provided, who spends how much time on what kinds of care, and how this compared to the caregivers’ preferences; and finally, (5) conditions of care recipients: we asked about the condition of the older care recipients (now and before), and at what point the family/household would turn to paid or external care support/help.

The survey respondents were checked for their role as the main caregiver based on baseline criteria, and samples checked for their national representativeness (We commissioned Angus Reid to conduct the survey using their panel community. The sample was representative by age and gender, but was slightly biased toward urban participants and people in English-speaking and southern Canada). Additionally, the sample was balanced on age and gender to reflect the profile of caregivers of elders over 65 years of age. The demographic profile of caregivers was determined by identifying caregivers within a national Omnibus survey with a representative sample of Canadian population that was conducted prior to the main survey. Interviews were conducted in English and French. For comparison purposes only, a sample of this size would yield a margin of error of ±3.1 percentage points 19 times out of 20. The survey yielded a total of 1000 responses. Of the 1000, we rejected 3 because of incomplete responses.

## 4. Results and Discussion

### 4.1. Basic Socio-Demographic Characteristics

Our survey shows a huge age spread amongst main caregivers, ranging from 18 to 88 years, with median and mean ages of 58 and 56, respectively. Unsurprisingly, women made up a slightly higher proportion of main caregivers compared to men (60% vs. 40%) (see Table 1 for a summary of socio-demographic characteristics). Most of the respondents were caring for either their parent (45% total: 42% of female caregivers; 49% for male caregivers) or their spouse (32% total: 34% female caregivers; 30% male caregivers), while a small proportion of the main caregivers were caring for their grandparents (7% total: 8% female caregivers; 5% male caregivers). 51.3% of respondents were living with the older people they were caring for, while 48.7% were living separately from their care recipients. Amongst those care recipients who were not living with the caregivers, about half (48.6%) were living on their own in their own house or apartment, 31.9% were living in a retirement home or long-term care institution; 11.1% were living in a house or an apartment with their spouse or partner, and 8.4% were living in assisted living accommodations. The four main reasons respondents gave for becoming the main caregivers for their spouses/partners or older relatives/friends were: (1) they loved their care recipient (46.9%); (2) their care recipients wanted them to take care of them (20.8%); (3) it was too expensive to hire a professional care worker (20.1%); and (4) they lived near their care recipient (20.0%) (Table 2). Most of the respondents in our survey were providing 1 to 5 h of care per day on weekdays: 34% and 33% of respondents spent 1–2 h and 3–5 h per day, respectively, providing care, and another 12% and 11% provided 6–10 h and more than 10 h of care a day, respectively. There were few differences between men and women, or amongst the different age cohorts in terms of the number of hours these caregivers spent providing care (Table 3). The four most common activities provided by these caregivers were: (1) taking their care recipients to hospital, the doctor’s office, bank, etc. (82%); (2) doing housework such as cleaning, tidying up, laundry, etc. (74%); (3) supporting them with public transport and other transportation (70%); and (4) preparing and eating meals and drinks (67%).

The socio-demographic characteristics of our survey respondents shared some similarities and differences with the results of previous Canadian national surveys, such as the 2018 General Social Survey (GSS)—Caregiving and Care Receiving (CCR) (Cycle 32) and 2021 Census. Our survey shows a similar gender ratio for female and male caregivers as the GSS survey (60% women:40% men compared with 54%:46%, respectively, for GSS). But caregivers in our survey reported spending fewer hours providing care compared with the caregivers in the GSS survey (67% spending 1–5 h, compared with 56% for GSS survey cohorts; only 11% of our survey cohorts spent more than 10 h per week, compared with 20% of GSS survey respondents spending more than 20 h per week). This may be because we only included people who identified as the main caregiver and they had to be providing care for someone who needed help with activities of daily living, whereas the GSS and CCR surveys included anyone who provided any care at all. The respondents in our survey were also more educated compared with the general Canadian population. Whereas three-quarters of our survey respondents (76% of women, 77% of men) had attained postsecondary/college or university degrees, the Census data suggest that 67% of adults in Canada aged 25 to 64 had a postsecondary certificate, diploma, or degree in 2021 [27].

### 4.2. Overall Health Impacts of Caregiving by Gender and Employment Status

About one-third of the respondents in our survey were fully retired. Among the non-retirees, 57% were carer–employees (CEs), the remainder being either unemployed or self-employed. The CEs rated their health status slightly higher than not-employed caregivers, with 38.91% of CEs rating their health status as “excellent” or “very good” compared with 33.64% of not-employed caregivers (Figure 1). These findings differ from those of previous research, which more often shows the negative impacts of combining caregiving and work on health, well-being, and workplace productivity for women and for CEs. Wu et al.’s (2023) comparative analysis of the two cycles of Canada’s General Social Survey (GSS 2012 and 2018), for example, shows that, overall, CEs report poorer general and mental health, higher stress, and lower and worsened life satisfaction compared with non-CEs (although according to Wu et al.’s data, GSS cycle 2018 found 86.32% of CEs and 92.79% of non-CEs rated their general health status as “Excellent/very good/good”. This is much more positive compared with our survey, where only 71.9% of CEs and 65.1% of non-CEs rated their health status as “Excellent/very good/good”).

Wu et al.’s research also found that women CEs were more likely to report poorer health compared with male CEs [6]. These findings are partially explained by the significant increase in the intensity of care for women CEs between 2012 and 2018. The health and well-being impacts of having to perform care and work are closely associated with both the intensity and the amount of care provided, and women are more likely to provide more care and more intense care for older people. Wu et al. thus concluded that the health and well-being outcomes of caregiving for CEs have remained consistent or worsened between 2012 and 2018, particularly for women. Indeed, other studies also identified intensity of care to be one of the most pernicious drivers of negative health and well-being outcomes for CEs [28]. Dugan et al. found that elder care demands had a deleterious impact on workers’ depressive symptoms, especially when such demands required five or more hours per week in addition to their work hours [9]. Our study found that men rated their health status slightly higher than women, with 38.58% of men rating their health status as “excellent” or “very good” compared with 35.44% of women (Figure 2).

Wu et al.’s [6] observation about the adverse effect of caregiving on employed caregivers supports other earlier Canadian CE studies [4,29,30,31]. In their analysis of the 2001 National Work–Life Conflict Study, Duxbury et al. argue that CEs, and in particular those who are providing eldercare, are at risk of experiencing caregiver strain compared with non-employed caregivers [29]. They show that CEs who are providing eldercare face multidimensional caregiver strains in the forms of physical strain (e.g., having to do heavy lifting and providing long hours of care, resulting in exhaustion and loss of sleep), financial strain (decline in income due to caregiving responsibilities, on the one hand, and on the other, increase in expenditures related to care), and emotional strain (increased stress, worries, sense of being overwhelmed, etc. resulting from role overload, uncertainty/worry, and empathy for the care recipient). Amongst these multidimensional caregiver strains, they specify emotional strain as the most stressful for CEs providing eldercare. As a result, these CEs are more likely to experience high levels of perceived stress and depression. Because women are more likely than men to give more intense and direct care to their older care recipients, women CEs are also more likely to report emotional strain. Duxbury et al. thus urge government and policymakers to provide supportive policies and programs to help reduce CEs’ caregiver strain. Possible supports they suggest include financial support, work flexibility, and other instrumental and community supports and respite care programs. Halinski et al. echoe Duxbury et al.’s study by noting that CEs were more likely to handle issues related to work and family by sacrificing personal time, rather than by lowering their workplace productivity, and this in turn, led to negative health outcomes [30].

Studies showing the negative impacts of double burden on caregivers are not limited to Canada. A large number of research and meta-analyses of caregiving studies in different national contexts also confirm worsening overall health and well-being for CEs [9,22,28,32]. For instance, in their meta-analysis of 84 caregiving studies spread across a wide range of contexts, Pinquat and Sorensen show significant differences between caregivers and non-caregivers in relation to depression, stress, self-efficacy, and general subjective well-being [15]. Bauer and Sousa-Poza’s meta-analysis also concurs with Pinquat and Sorensen’s finding [33]. They argue that, even though caregiving can create psychological uplifts, these positive effects are outweighed by negative consequences, particularly for women and for spousal caregivers. Moreover, they pointed out that employment, family, and health have mutual and reinforcing effects on each other that can lead to declines in health and well-being outcomes.

Our survey findings agree with previous findings and analyses about the gendered dimensions of caregiving, with women reporting their overall health status being lower than that of male caregivers, though the difference is not very large. However, our findings diverge from those of previous research in that our survey indicates better self-reported health status for CEs than caregivers who are not employed. There are several possible explanations for our unexpected findings. One explanation may be that our comparator groups were different from those of other studies. Many Canadian studies of CEs employ large national surveys, such as GSS, to assess health and well-being impacts of unpaid caregiving by comparing CEs with non-CEs (i.e., employees without caregiving obligations). Our study, however, is specific to eldercare caregivers, and we compare CEs with not-employed caregivers. It is therefore possible that, while CEs in our survey may have reported more positive self-reported health status compared with non-employed caregivers, they may still have had lower self-reported health status compared with employees who are not providing care to elderly people. Nevertheless, that CEs in our survey appeared to have better self-reported health compared to non-employed caregivers is puzzling given that CEs will likely experience work–family conflict and family–work conflict, whereas non-CEs would not face such tension. This contradicts Duxbury et al.’s findings, which compared CEs providing eldercare with non-employed eldercare caregivers [29]. A more in-depth analysis comparing these two groups of caregivers would be necessary to understand the finding. To gain a better and more detailed understanding of these caregivers’ perspectives and lived experiences of providing care for their older family members, and to find out in more detail how combining work and care might affect caregivers’ health and well-being, we conducted a followed up in-depth interview survey with 57 respondents in the spring of 2023. We are currently analyzing the interview data.

A second explanation for our results may be that our data for self-reported health status by gender and by employment may have masked the difference in subjective health status of female CEs and non-CEs, and between male and female CEs because they are aggregated data. We are currently running the regressions to gain a better picture of the relationships between gender and employment status on self-rated health status outcomes.

Another possible explanation for the better subjective health outcomes for CEs in our survey may be that those who are working are less likely to have financial stress. Adding to this, and contrary to expectations, work and other non-caregiving activities might also serve as a break or an escape from caregiving, instead of further contributing to caregiver burden. A few studies suggest that paid work in some cases may benefit caregivers’ well-being [7]. In their earlier study, Bainbridge et al. used national survey data to assess whether hours of work were associated with caregiver stress outcomes [8]. Their findings suggest that paid employment may serve as a resource, rather than a detriment, buffering the negative effects of caregiving by defraying the high cost of caregiving. Moreover, they found that CEs’ involvement in a work role had neither an adverse nor a beneficial impact on caregiver stress outcomes. Rather, disability type did interact with weekly hours of work to influence stress outcomes, such that, while the stress outcomes of caregivers providing care for people with non-mental disabilities were unaffected by the caregivers’ time at work, those caring for a recipient with a mental disability benefited from outside work, experiencing significantly fewer stress outcomes when they spent more time at work. They thus argue that, depending on the care recipient’s disability type, engaging in employment may be in fact beneficial for caregivers.

In their study examining the nexus between paid work and caregiving for Australia’s baby boomer cohort, O’Loughlin et al. found that, compared with non-caregivers, caregivers reported lower workforce participation (45.8% versus 54.7% for non-carers), as well as poorer health, more mobility difficulties, lower quality of life, and lower self-rated socio-economic status [13]. However, amongst the caregivers, those who were working tended to have fewer mobility difficulties, better self-rated health, and higher socio-economic status than those not working (This however may be a reflection of the gendered nature of outcomes as in their study male carers were more likely than female carers to be in full-time or no paid work).

Finally, a fourth explanation for the better subjective health outcome for CEs in our survey may be the timing of the survey. Previous national surveys of caregiving were almost all conducted before 2020, and subsequent analyses of caregivers’ health outcomes were conducted using those surveys. Our survey, however, was conducted in 2022, during the tail end of the COVID-19 pandemic. As many employees in Canada were working virtually or semi-virtually and/or were working under much more flexible working arrangements during this time, these non-traditional work arrangements or simply the COVID-19 context may have had some moderating effect on employees’ family–work or work–family role conflicts that resulted in better subjective sense of health and well-being. Our online survey did not ask questions related to the COVID-19 context. Given that our follow-up interview survey was conducted in the spring of 2023, shortly after the COVID pandemic, we were able to hear the interviewees’ experience of caregiving during COVID. We are currently analyzing the follow-up interview data to gain a better understanding of this.

### 4.3. Mental Health Impacts of Caregiving on Carer-Employees

Beyond overall self-rated health status, our national online survey found no significant differences between CEs and non-CEs in terms of other mental and physical health outcomes, such as tiredness, depression, anxiety, difficulty sleeping, physical strain, and worsening health. Again, these findings differ somewhat from those of most other previous research. For example, using data from the 2012 Canadian General Social Survey on Caregiving and Care Receiving, Li and Lee showed poorer mental health outcomes for CEs compared with non-employed caregivers [31]. They found that these negative mental health outcomes, including lower self-rated mental health and higher life stress levels, are significantly associated with employment adjustment. Moreover, they argue that, on the one hand, family–work role conflict serves as a mediator; on the other hand, workplace support serves as a moderator in this relationship. Accordingly, family–work role conflict mediates between employment adjustment and mental health outcomes, and that while this mediating effect is significant at all levels of workplace support, it weakens as workplace support increases. In other words, when employed family caregivers make employment adjustments due to family caregiving responsibilities, participants with a higher level of workplace support are less likely to experience worse mental health due to perceived family–work role conflict, largely because positive workplace support enables employed family caregivers more control over their working hours or schedules, thus providing better balance work and family.

As discussed in the previous section on self-rated overall health outcomes for CEs, the positive mental health outcomes for CEs in our survey may be a result of data aggregation and the timing of the survey. To gain a better understanding of our survey result we will undertake regression analyses of the relationships between mental health outcomes and gender and employment status, and between gender and employment status. Furthermore, it would be useful to conduct follow-up surveys to assess the causal effects of caregiving and work on the mental health of CEs over time. Ervin et al., for example, specifically addressed this issue in their longitudinal study of unpaid caregiving and mental health among working-age adults in high-income OECD countries [32]. Consistent with the findings of previous reviews, they reveal a negative association between informal unpaid care and mental health among adults of working age.

### 4.4. Pandemic—A Reminder of Gendered Outcomes for Caregivers

Our survey is one of few studies conducted during the COVID-19 pandemic in Canada. There is no doubt that the pandemic was an unprecedented negative health shock that placed additional strain on households already facing challenges in managing time and resources devoted to the care of elders, children, or the ill. Additionally, a significant number of individuals found themselves unexpectedly assuming caregiving roles they had not held before, while simultaneously managing these responsibilities alongside their paid employment duties [34]. Most studies of caregiving during the COVID-19 pandemic show overall negative health and psychological well-being for caregivers during this period. For example, using data from the UK Household Longitudinal Study, Madia et al. found that the pandemic had a greater impact on the mental health of caregivers compared with the general population throughout the entire period [34]. A growing body of literature on the impact of COVID-19 upon caregivers also showed that caregivers experienced increased stress due to factors such as social isolation, reduced support networks, lack of information, added challenges in daily caregiving activities, and concerns for their own well-being. In their scoping review of this literature, Muldrew et al. argued that, during COVID-19, caregivers experienced a decline in psychological well-being and personal health, with increased practical and logistical concerns, and withdrawal or suspension of the care recipients’ key support services [35].

Other studies also show that the pandemic was doubly disruptive for carer employees. Truskinovsky et al.’s study on a national cohort of US adults aged 55 years and older, for example, found that family caregivers experienced worse mental and physical health than non-caregivers during the first wave of the COVID-19 pandemic [36]. Caregivers in the labor force who experienced disrupted care arrangements, especially those providing more care, were also likely to experience disrupted employment, particularly in the forms of job loss, furlough, or a work-from-home transition. Unfortunately, their study was focused on whether individuals who began providing care during the pandemic reduced their work hours due to increased caregiving responsibilities. As they did not investigate the dual impact of employment and caregiving on caregivers’ mental and physical health, it is unclear how the impact of the pandemic on CEs compared with non-CEs.

This aspect was addressed by Wister et al. in their study utilizing data from the Canadian Longitudinal Study on Aging (CLSA) [37]. They found that participants who were younger, unmarried, employed, immigrants, and living in an urban area reported higher levels of depressive symptoms and/or anxiety during the pandemic. Specifically, employed participants experienced a higher level of depressive symptoms and anxiety compared with those not in the labor market. They also highlight that female caregivers reported greater depressive symptoms and anxiety than their male counterparts. The study by Madia et al. corroborate the finding that the gender gap in mental health widened during the pandemic, with women more likely to report mental health issues [34].

The challenges of the pandemic proved particularly daunting for women in heterosexual relationships, primarily due to the higher likelihood of traditional gender role distribution (i.e., male/provider and female/caregiver). The challenge of accommodating remote work alongside household duties and full-time caregiving responsibilities worsened pre-existing gender inequalities, with women being the “default” caregiver most of the time, despite also being employed [38]. Stefanova et al., in their study of 240 heterosexual individuals with or without caregiving responsibilities who lived with a partner and worked from home during the pandemic, found a significant gender imbalance, such that female caregivers spent significantly less time on work compared with the other groups and significantly more time on caregiving compared to male caregivers during the lockdown [38]. Their sample, spread across the US, UK, and Ireland, showed a significant direct effect of caregiving on career outcomes for women, such that the more caregiving women performed during the lockdown relative to other tasks, the more negative their self-reported career outcomes were. Overall, their study show that the gender imbalance in distributions of caregiving duties during the pandemic was strongly associated with negative personal and professional outcomes for women who were caregivers.

The fact that our survey, taken during the COVID pandemic period, found better self-reported general health status for CEs compared with non-CEs, and no significant differences between CEs and non-CEs in terms of other health outcomes such as tiredness, depression, anxiety, difficulty sleeping, physical strain, and worsening health is not only puzzling, but also seems to contradict most existing studies. One possible explanation may be that our survey was focused on eldercare, rather than childcare or caregiving in general. Most studies of caregiving stress during the COVID-19 pandemic have focused on childcare, which seem to show that working parents were most affected by the changes in work patterns resulting from the COVID restrictions. Although Stefanova et al.’s study did not specify the breakdown of caregivers caring for a child and older people, their sample’s younger-age cohort seemed to suggest that a large proportion of caregivers were providing childcare, rather than eldercare [38]. Indeed, much less is known about the mental and physical health impacts of caregiving for CEs providing eldercare compared with childcare during the COVID-19 pandemic. It may be that the pandemic might have affected caregivers, both CEs and non-CEs, caring for children and young adults more than those caring for older people. We also suspect that in Canada, while the COVID restrictions may have had added extra care burden on CEs’ providing eldercare, particularly if they are co-residing with their care recipient, a shift to more flexible work hours and access to financial support, such as CERB may have helped mitigate some of the adverse impacts of the pandemic. Again, analyses of our follow-up interview survey will be able to show in more detail the CEs’ caregiving dynamics in relation to the COVID-19 pandemic.

## 5. Policy and Program Implications

The findings of this study have important implications for gender, care, and work–family policies and programs. The differential impacts of unpaid caregiving between women and men, consistent with many other previous studies, and between CEs and non-CEs need to be addressed by policymakers, employers, co-workers, and communities. Evidence shows that, across the globe, unpaid caregivers provide huge amounts of invaluable care and support for their families, friends, and community members, yet much of these social and economic contributions remain invisible and unaccounted for. As well, they are ignored in the national GDP and formal economic thinking. It is therefore important for policymakers, employers, and other social and economic actors to recognize the value of unpaid caregiving, and to develop meaningful policies and programs that will acknowledge its contributions and reduce the care burden on caregivers. As our population continues to age, this should be an important policy priority for governments, employers, and community organizations. In considering the importance of unpaid caregiving, it is also crucial that policies and programs recognize the unequal distributions of caregiving and care work between women and men, and amongst different groups of people—employees, workers, and non-employees, and people in different socio-economic classes and age groups—and unequal impacts of caregiving on different groups of people. We therefore must apply an intersectionality lens to develop gender-sensitive, inclusive, and family friendly policies and programs that will address the different and diverse needs of caregivers. This would include, among other actions: (1) more publicly provided accessible and quality care services for older people, children, and people with disabilities; (2) support for caregivers, such as respite services, community programs, and education for caregivers and their care recipients, financial support for caregivers and their families; (3) generous care leaves; (4) flexible work place policies and practices; (5) more employer sensitivity and understanding towards the needs and challenges of workers with care responsibilities in hiring and promotion practices; and (6) better integration of social and healthcare systems. To achieve these, it will be imperative to allocate more resources and funding to care and the social infrastructure needed to support caregivers.

Future research on CEs and caregiving should investigate more precisely the mechanisms behind the differences in health and well-being outcomes of female and male caregivers and between CEs and non-CEs, particularly under crisis contexts such as the COVID-19 pandemic. For example, how do social and cultural norms about gender and caregiving shape men and women’s decision to become main caregivers and the way they provide care? Are there differences in caregiving patterns and health and well-being outcomes between younger and older caregivers, and between younger and older CEs and non-CEs? In what ways do changes in employer behavior or work flexibility mitigate or reinforce caregivers’ care burden, and are they the same or different between women and men? What effects do financial supports, such as CERB or paid care leaves, have on caregivers’ health and well-being? Are there differences in health and well-being outcomes of caregivers living in urban and rural areas, and why? All these studies will benefit from more rigorous analyses and qualitative research.

## 6. Conclusions

This paper discussed the preliminary findings of the Canadian survey on unpaid caregiving for older people conducted in 2022. Ours is one of very few surveys examining the impact of unpaid caregiving during the COVID-19 pandemic period. The descriptive data show mixed results for health and well-being outcomes for caregivers. Whereas women caregivers had worse self-rated health and well-being compared with male caregivers, concurring with many previous studies, the outcomes were more positive for CEs compared with non-CEs, which noticeably differed from many of the previous studies. I have raised potential explanations for these similarities and differences, including the lack of data disaggregation, the need for more in-depth regression analyses and additional qualitative interview survey, and the timing of our survey.

This study advances our understanding of the multiple effects and dimensions of unpaid caregiving for older people on women and carer–employees, and raises an important policy and program agenda item for policymakers, employers, and other social and economic actors to consider. The interplay between caregiving responsibilities and work demands can generate significant stress, impacting the caregiver’s quality of life and mental health. However, it is also crucial to recognize that a caregiver’s declining mental health can also have adverse effects on the well-being of the care recipient [34]. This underscores the interconnectedness of caregiver and care recipient outcomes, highlighting that the well-being of CEs holds implications not only for themselves, but also for their care recipients, employers, and the healthcare system at large [39].

The findings from this study, and from existing literature, highlight the urgent need to address and mitigate the health risks associated with caregiving among CEs, particularly given the continuing aging of the population and therefore ongoing demand for care. The disproportionate burden of caregiving placed on women presents a pressing and urgent matter for policy consideration and societal action. Women, particularly those juggling caregiving responsibilities with employment and professional careers, often face compounded challenges that can detrimentally affect their mental health, economic stability, and overall well-being. Reducing this disproportionate caregiving load is essential for promoting gender equality, supporting women’s workforce participation, and ensuring the sustainability of caregiving arrangements for families, communities, and society.

## 7. Limitations

As this paper is based on an analysis of descriptive data from the national survey, there are many limitations. Given the lack of data disaggregation and regression analyses, I was not able to provide clear causal relationships and associations between the health and well-being outcomes of male and female caregivers and between CEs and non-CEs. Rather, this paper was only able to compare some of the global health and well-being outcomes of caregiving between men and women and CEs and non-CEs. Moreover, this paper was limited by the lack of qualitative data to provide more in-depth and nuanced information. The regression analyses of the survey are currently in progress. Also, following the national survey, we have conducted interviews with 57 of the survey respondents in 2023. We are currently analyzing those data as well.

## Figures and Tables

**Figure 1 ijerph-21-01611-f001:**
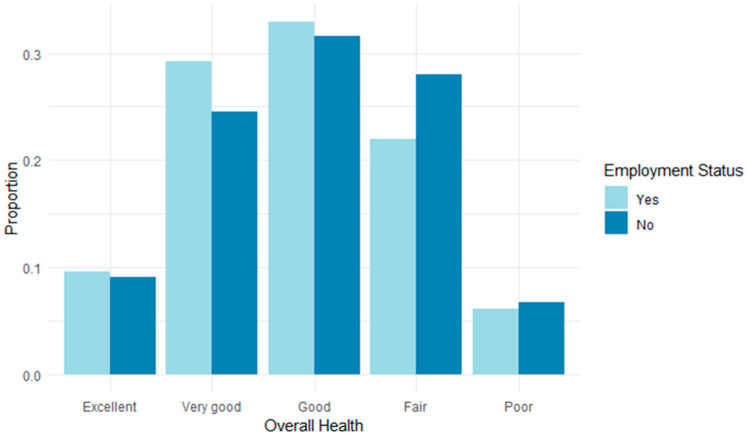
Self-rated health status by employment status.

**Figure 2 ijerph-21-01611-f002:**
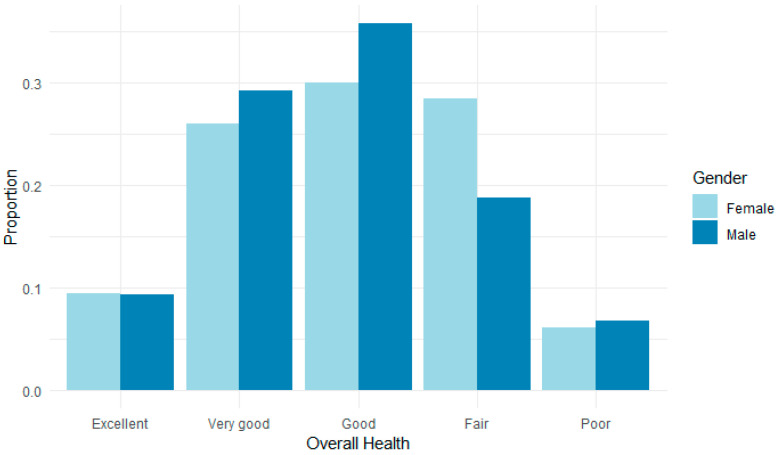
Self-rated health status by gender.

**Table 1 ijerph-21-01611-t001:** Socio-demographic characteristics.

	Female (%)	Male (%)
Gender	(total number: 601)	(total number: 395)
	60%	40%
Age		
18–34	7	12
35–54	26	29
55+	66	59
Marrital Status		
Single/never married	16	19
Married/Common Law	64	71
Separated	5	3
Divorced	11	5
Widowed	4	2
Employment		
Employed	46	54
On leave (maternity, short or long-term disability, etc.)	7	5
Fully retired	35	33
Unemployed, looking for work	4	3
Unemployed, not looking for work	8	6
Education		
Highschool or less	25	24
Postsecondary/college	33	34
University+	43	43
Household Income		
<CAD 25 K	8	8
CAD 25 K to less than 50 K	16	14
CAD 50 K to less than 100K	34	30
CAD 100 K to less than 150K	16	25
CAD 150 K to less than 200K	7	10
>CAD 200 K	6	7
Do Not Know/rather not say	13	6
Citizenship		
Canadian citizen (born in Canada)	84	82
Canadian citizen (not born in Canada)	12	14
Permanent resident	1	3
Refugee	1	1
Other	1	1
Ethnicity		
White	72	74
Other	21	24
Prefer not to say	2	3
Visible Minority		
Yes	18	25
No	82	75
Care receiver’s relationship to respondent (living together)	(Base number: 297)	(Base number: 214)
spouse/common-law partner	34	30
paternal grandparent	3	3
maternal grandparent	5	2
parent	42	49
Parent of spouse/common-law partner	6	9
sibling	2	1
Sibling of spouse/common-law partner	*	
Unmarried child	*	
Married child/child who is living common-law	*	*
Spouse of common-law of child	*	1
Other relative	3	3
Neighbor		*
Non-relative (friend/acquaintance)	3	1
Care receiver’s relationship to respondent (not living together)	(base number: 305)	(base number: 183)
spouse/common-law partner	2	1
paternal grandparent	2	5
maternal grandparent	1	5
parent	75	64
parent of spouse/common-law partner	6	10
sibling	2	3
sibling of spouse/common-law partner	1	1
married child/child who is living common law	*	1
other relatives	6	3
neighbor	1	4
non-relative (friend/acquaintance)	4	4
Living Arrangement (where does the care receiver live?)	(All respondent: 602)	(All respondent: 397)
with main caregiver (respondent)	49	54
His/her own house/apartment	24	22
In a house/apartment with spouse/partner or with other family members	6	4
In a retirement home or long-term care facility/home	16	15
Other assisted living accommodation	4	5
Net: In a retirement home/LTC/Assisted living	20	20
Use of paid public or private external care services/institutions for the care of the recipient)	(Base number: 602)	(Base number: 397)
Yes	56	55
No	44	45

* no answer obtained.

**Table 2 ijerph-21-01611-t002:** Reason for becoming the main caregiver.

Reason for becoming the caregiver (cumulative % top 2 reasons)	I am the only family member of the older person	9.5%
	I have been living with the older person	15.9%
	I live nearest to the older person	20.0%
	I (or my spouse) am the first child of the older person	9.1%
	The older person wants me to take care of him/her	20.8%
	All other family members work, so I am the only available person to take care of the older person	11.6%
	I care for the older person to not bother other family members	5.2%
	Because I love him/her	46.9%
	Because I feel most comfortable giving care myself	9.0%
	Because I am able to provide the best care for him/her	19.7%
	Because it is too expensive to hire a professional care worker	20.1%
	Because I cannot find professional care service for him/her	6.7%
	Other (specify)	8.1%

**Table 3 ijerph-21-01611-t003:** Average amount of hours spent providing care per day during weekdays.

	Total	Gender		Age		
		Male	Female	18–34	35–54	55+
	(A)	(B)	(C)	(D)	(E)	(F)
BASE: All Respondents	1000	397	602	92	274	634
Less than 1 h/day	94	40	53	15	25	54
	9%	10%	9%	16%	9%	9%
1–2 h/day	343	150	193	31	99	213
	34%	38%	32%	34%	36%	34%
3–5 h/day	334	128	206	31	95	208
	33%	32%	34%	34%	35%	33%
6–10 h/day	120	41	79	9	34	77
	12%	10%	13%	10%	12%	12%
More than 10 h/day	109	38	71	6	21	82
	11%	10%	12%	7%	8%	13%

## Data Availability

The original contributions presented in this study are included in the article, further inquiries can be directed to the corresponding author.

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
