# Peer review of "Mental and Physical Wellbeing of Carer–Employees in Canada"

_ijerph, 2024, doi:10.3390/ijerph21121611_

Round 1

Reviewer 1 Report

Comments and Suggestions for Authors

Brief Summary: The aim of the study is to holistically understand caregiving and the impact of unpaid caregiving socially, economically and health wise. The relevance of the study is well established and contemporary, contributing to the dialogue on carer-employee demands and outcomes for the individual, the person in receipt of care, and society. Framing the study within theoretical explanations is useful and recommendations are always welcome.

Article: 

Can the author comment on the data collection and whether the approach (commissioning Angus Reid and accessing their panel community) may have excluded certain cohorts of the population and may explain some of the findings.

Because the survey was conducted during the Covid-19 pandemic, can the author comment on whether a consideration was giving to exploring, at some level, pre and post pandemic experiences and potential impacts. Perhaps this is discussed in the interviews currently under analysis?

As stated, situating papers within theoretical frameworks is useful, however it is not clear in this paper which if any guided the study, can this be clarified?

Making recommendations that have cost implications can be controversial, how these will be funded is usually a caveat, and is not addressed here.

Overall, the paper is well-presented, tables are clear, legible and easy to understand. Sufficient detail has been included in the methods to enable further study in the area. The implications for future research are well considered. 
Generally, references included are relevant, though some are quite dated and would benefit from review if possible.

I am a little confused as the paper states that the survey was part of the SSHRC funded global partnership research 145 project, Care Economies in Context: towards sustainable social and economic development. Can the author clarify any co-authors and why there is no funding associated with this paper.

Author Response

Thank you very much for your careful review of the article. I found them very useful in helping revise my article. I have done some small typo corrections and grammatical edits, and highlighted the revisions I made in response to the comments from the reviewers in the revised article. Below please see my response to your comments and feedback in blue. While I tried my best to address reviewers' concerns for more information, I am also aware of the word limits for the article. So I tried my best to add more information within the word limit.

Again, thank you very much.

with warm regards,

Ito Peng

-------------------------

Reviewer #1:

Comments and Suggestions for Authors

Brief Summary: The aim of the study is to holistically understand caregiving and the impact of unpaid caregiving socially, economically and health wise. The relevance of the study is well established and contemporary, contributing to the dialogue on carer-employee demands and outcomes for the individual, the person in receipt of care, and society. Framing the study within theoretical explanations is useful and recommendations are always welcome.

Thank you very much for your feedback. I appreciate this very much.

Article: 

Can the author comment on the data collection and whether the approach (commissioning Angus Reid and accessing their panel community) may have excluded certain cohorts of the population and may explain some of the findings.

I have made revisions in Data and Methods section on p. 4, and added an explanation on the limitations of the data collection in the footnote.

Because the survey was conducted during the Covid-19 pandemic, can the author comment on whether a consideration was giving to exploring, at some level, pre and post pandemic experiences and potential impacts. Perhaps this is discussed in the interviews currently under analysis?

I added a discussion on this at the discussion section of the paper.

As stated, situating papers within theoretical frameworks is useful, however it is not clear in this paper which if any guided the study, can this be clarified?

I applied the general and shared understanding from previous research about adverse health and mental health effects family caregiving on employed workers. As this is a preliminary and descriptive summary overview of our research finding, I wanted to be careful in not drawing direct causal links between our survey outcomes and specific variables. I find the Effort-Recovery and Pearlin's Stress Process models compelling, but I decided to hold back on explaining in depth these models until we complete a more indepth analysis.

Making recommendations that have cost implications can be controversial, how these will be funded is usually a caveat, and is not addressed here.

I have added a statement about a need to allocate dedicated resource and funding to achieve the policy recommendations in the policy implication section.

I am a little confused as the paper states that the survey was part of the SSHRC funded global partnership research 145 project, Care Economies in Context: towards sustainable social and economic development. Can the author clarify any co-authors and why there is no funding associated with this paper.

I have clarified this in the Funding section of the article.

Reviewer 2 Report

Comments and Suggestions for Authors

Abstract:

Please describe the method and the sample

Do not write I conclude, but describe the most important policy for care-givers

1. Introduction:

Clear the aim of the paper and write a clear research question. The last sentence (p2, line39) seems that you are only discussing the results. Describe more detailed the paper and its chapters.

2. Theoretical/ Conceptual Explanations

Please describe the results of study, also with numbers, how many persons of the samples are affected of stress?

3. Data and Methods

Wich university approved the ethic questions?

When were the data ciollected?

Which tool was used?

Clear the questionaire and create a table with sample questions and the sources - compared with  the results the Questionaire is not clear

How did you try to get the right sample of working care-givers?

At Line 333 the Covid-19 pandemic is the first time mentioned. This has an impact on the paper and must be mentioned earlier, at least in the method

Points of the discussion is already discussed in the results (like the
Covid-19-pandemic). This must be separetd or clarify with the publisher,
if this must be separated or can bei in one chapter

Sharpen the conclusion and limitation

Please control if all abbriviation are clear

Author Response

Clear the aim of the paper and write a clear research question. The last sentence (p2, line39) seems that you are only discussing the results. Describe more detailed the paper and its chapters.

I have added a sentence to give a bit more detail about the paper.

  1. Theoretical/ Conceptual Explanations

Please describe the results of study, also with numbers, how many persons of the samples are affected of stress?

I tried to do this by showing describing the results in sentences and in figures/graphs.

  1. Data and Methods

Wich university approved the ethic questions?

Thank you, I have addressed this in the article.

When were the data ciollected?

I have also addressed this in the article.

Clear the questionaire and create a table with sample questions and the sources - compared with  the results the Questionaire is not clear

The actual questionnaire is over 7 pages long and unfortunately I cannot add this onto the article within the word limit.

How did you try to get the right sample of working care-givers?

We tried our best to get a nationally representative sample of caregivers as the survey as about unpaid caregiving for older people. The working caregivers are the subset of the total samples of unpaid caregivers providing care to older people.

At Line 333 the Covid-19 pandemic is the first time mentioned. This has an impact on the paper and must be mentioned earlier, at least in the method

Thank you for this. I have tried to address this in my revision.

Points of the discussion is already discussed in the results (like the
Covid-19-pandemic). This must be separetd or clarify with the publisher,
if this must be separated or can bei in one chapter
Thank you for the suggestion.

Sharpen the conclusion and limitation

Thank you for the suggestion.

Please control if all abbriviation are clear

Thank you for the suggestion. I have tried to keep the abbreviations clear.

Reviewer 3 Report

Comments and Suggestions for Authors

Article  Mental and physical wellbeing of carer-employees in Canada approaches very important topic, which is relevant to the Journal. Article is very well-written and structured. There are only minor issues that needs attention.

-Line 46-46: Author is stating that women provide more intence care than men. Could author elaborate in which sense more intence? What does that mean in practice?

-there is great description of previous studies.

- paragraph from line 90 is very long, is it possible to divide it in two?

-From time to time care studies (in elderly care and disability care) have been blamed for being negative and producing negative results, because of the preliminary hypothesis of researchers that care is a burden and reframing questions from this perspective. This means that focus of the studies could be moved also to positive factors that care produces for carer, to coping mechanisms of carer etc. I think author has brought up some results from this perspective from previous research, but maybe this idea could be considered in the end of manuscript when describing future researh needs. 

-Results of the research are clearly described.

-The results of the study are important and interesting. Study found better self-reported general health status for CEs compared to non-CEs, and no significant differences between CEs and non-CEs in terms of other health outcomes such as tiredness, depression, anxiety, difficulty sleeping, physical strain and worsening health, which seem to contradict most existing studies.  Author elaborate explanations for this result from different perspectives. One explanation which is used in research on COVID-19 and families with children is that during COVID many obligations (travelling to work, hobbies etc.) were reduced so people (at leats those in priviledged positions) could have more time for themselves and for rest, which supported their well-being. Also there is studies about coping mechanism that was used during COVID-19 and those included acceptance, greatfulnes and family-centered activities. This could mean that peoples attitude towards caring was more positive than before pandemic, because situation was so unpredictable, there were limited social contact and risks of COVID-19 to elderly was constantly on the news, which could create fear of loosing loved ones and appreciation of ones life situation. Maybe author could check out if this perspective is relevant to the study.

I thank for possiility to review such a great article and wish author good luck in finalizing stage!

Author Response

Article:  "Mental and physical wellbeing of carer-employees in Canada” approaches very important topic, which is relevant to the Journal. Article is very well-written and structured. There are only minor issues that needs attention.

Thank you for your feedback. I appreciate this very much.

-Line 46-46: Author is stating that women provide more intence care than men. Could author elaborate in which sense more intence? What does that mean in practice?

Thank you for your comment. I have added more information about what I mean by more intense.

-there is great description of previous studies.

Thank you for your comment.

- paragraph from line 90 is very long, is it possible to divide it in two?

Thank you for your suggestion. I have divided this paragraph into two.

-From time to time care studies (in elderly care and disability care) have been blamed for being negative and producing negative results, because of the preliminary hypothesis of researchers that care is a burden and reframing questions from this perspective. This means that focus of the studies could be moved also to positive factors that care produces for carer, to coping mechanisms of carer etc. I think author has brought up some results from this perspective from previous research, but maybe this idea could be considered in the end of manuscript when describing future researh needs. 

I think this is a very interesting and important point. I agree with the reviewer that care and carework are very complex and nuanced activity and relationship, and as such, while much of this work is hard and difficult, it could also be highly positive and rewarding, depending on the context. Our surveys (we have done large-n national surveys of paid and unpaid care for children and older people, and follow-up indepth interview surveys with caregivers) show that while caregiving may be challenging and negative, there are also positive dimensions to is as well. I think it is important not to only focus on the negative aspect of caregiving.

-Results of the research are clearly described.

Thank you.

-The results of the study are important and interesting. Study found better self-reported general health status for CEs compared to non-CEs, and no significant differences between CEs and non-CEs in terms of other health outcomes such as tiredness, depression, anxiety, difficulty sleeping, physical strain and worsening health, which seem to contradict most existing studies.  Author elaborate explanations for this result from different perspectives. One explanation which is used in research on COVID-19 and families with children is that during COVID many obligations (travelling to work, hobbies etc.) were reduced so people (at leats those in priviledged positions) could have more time for themselves and for rest, which supported their well-being. Also there is studies about coping mechanism that was used during COVID-19 and those included acceptance, greatfulnes and family-centered activities. This could mean that peoples attitude towards caring was more positive than before pandemic, because situation was so unpredictable, there were limited social contact and risks of COVID-19 to elderly was constantly on the news, which could create fear of loosing loved ones and appreciation of ones life situation. Maybe author could check out if this perspective is relevant to the study.

Thank you very much for this comment. I discussed a bit more on this in the discussion section. We are currently analyzing interview surveys that were done after COVID-19. And we are getting a much better understanding of the impact of COVID-19 on caregivers of children and older people.